# High-resolution 3D Maps of Left Atrial Displacements using an Unsupervised Image Registration Neural Network

**Christoforos Galazis**                     C.GALAZIS20@IMPERIAL.AC.UK
**Anil Anthony Bharath**                     A.BHARATH@IMPERIAL.AC.UK
**Marta Varela**                             MARTA.VARELA@IMPERIAL.AC.UK
*Imperial College London, UK*

## Abstract

Functional analysis of the left atrium (LA) plays an increasingly important role in the prognosis and diagnosis of cardiovascular diseases. Echocardiography-based measurements of LA dimensions and strains are useful biomarkers, but they provide an incomplete picture of atrial deformations. High-resolution dynamic magnetic resonance images (Cine MRI) offer the opportunity to examine LA motion and deformation in 3D, at higher spatial resolution and with full LA coverage. However, there are no dedicated tools to automatically characterise LA motion in 3D. Thus, we propose a tool that automatically segments the LA and extracts the displacement fields across the cardiac cycle. The pipeline is able to accurately track the LA wall across the cardiac cycle with an average Hausdorff distance of $2.51 \pm 1.3\ mm$ and Dice score of $0.96 \pm 0.02$.

**Keywords:** Left Atrial, Image Registration Neural Network, Displacement Field Vector.

## 1. Introduction

The analysis of the anatomy and function of the left atrium (LA) is becoming more important for the prognosis and diagnosis of cardiac conditions such as atrial fibrillation (AF) or heart failure (HF) (Hoit, 2017; Peters et al., 2021). Structural characteristics of the LA are established atrial disease biomarkers (Varela et al., 2017) and analysis of LA deformations has been explored using speckle-tracking echocardiography (Smiseth et al., 2022). These biomarkers are typically obtained for a single LA view and spatial averages across LA wall regions. Spatiotemporal 3D maps of LA deformation are expected to provide more specific signatures of LA pathology, with greater diagnostic and prognostic value, as has been shown for the left ventricle (LV) (Duchateau et al., 2020). However, there are currently no publicly available MRI datasets or adequate image analysis tools to extract high-resolution displacement field vector (DFV) maps of the whole LA.

In this paper, we use a novel high-resolution Cine MRI protocol designed specifically for the LA. These Cine MRI offer information about the LA at higher spatial resolution than images of any other existing database. However, given that only a small number of subjects have been imaged with this protocol, we develop and utilize methods for limited number of training images.

**Aim** We propose the following pipeline to automatically obtain high-resolution 3D DFVs of the LA: 1) A few-shot segmentation network (LA-SNet) of the LA across the cardiac cycle to guide the registration; 2) Extraction of the LA segmentation contour and dilation; 3) An automatic subject-by-subject image registration of the LA contour image (LA-DNet).

## 2. Methods

### 2.1. Data

We use 3D LA Cine MRI bSSFP scans acquired using a novel acquisition protocol (Varela et al., 2020). In summary, they were acquired in a single breath-hold, with resolution of $1.72 \times 1.72 \times 2.00 \ mm^3$ and 20 phases across the cardiac cycle. Phase 0 corresponds to cardiac end diastole (smallest LA volume). As proof of concept, we analyse images from six subjects: three healthy volunteers and three subjects with suspected cardiovascular disease.

### 2.2. Preprocessing

The images are cropped to a size of $96 \times 96 \times 36$ voxels, centered at the LA. Additionally, they are translated such that the LA centroid is stationary across the cardiac cycle and their intensity is min-max normalized. We manually segment the LA across the entire cardiac cycle to use as ground truth. From the segmented data, the contour is extracted and dilated using a 2 voxel radius spherical structure, which is used to mask the images.

### 2.3. Model

Details of LA-SNet and LA-DNet are in Figure 1, which their parameters have been experimentally selected. They share the same architecture that is based on a 3D U-Net (Ronneberger et al., 2015). The models incorporate squeeze and excitation blocks (Hu et al., 2018), which were already applied to LV MRI segmentation and image registration (Galazis et al., 2022). LA-DNet also utilizes a spatial transformer (Jaderberg et al., 2015) to obtain the DFV in an unsupervised way. The DFV is smoothed using a bending energy regularizer (Rueckert et al., 1999). LA-SNet is trained on the augmented whole LA images on cardiac phases 0, 8, and 15 and predicts the respective LA segmentation. LA-DNet takes the two contour masked images (moving: cardiac phase 0; fixed: cardiac phase [0-19]) to generate a displacement field that resamples the moving to the target image.

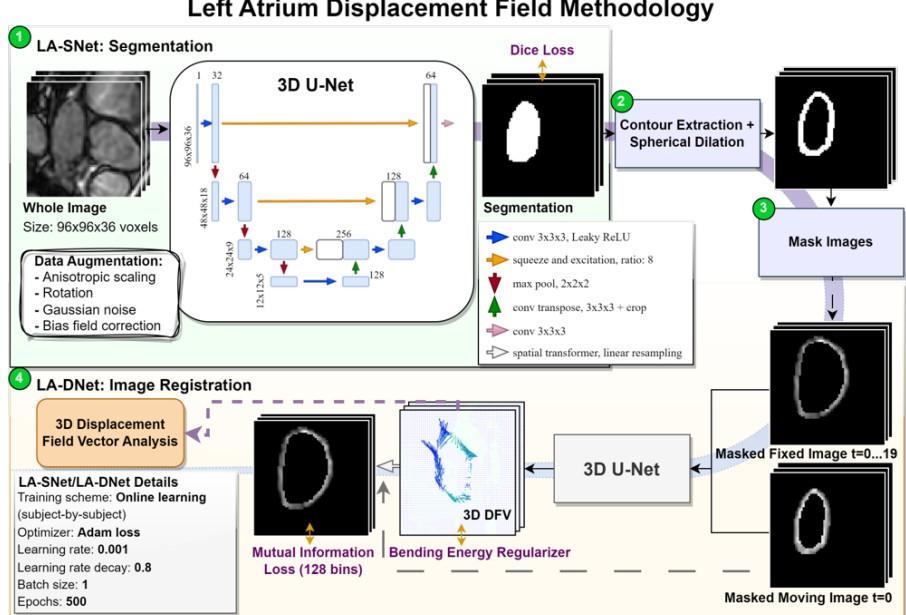

Figure 1: The proposed pipeline to extract high-resolution LA displacement field maps.

## 3. Results

LA-SNet can accurately segment the LA across the cardiac cycle, with an average Hausdorff distance (HD) of $3.03 \pm 1.12$ $mm$ and Dice score (DS) of $0.95 \pm 0.02$. Similarly, LA-DNet is able to accurately track the LA wall across the cycle (see Figure 2). The LA segmentations obtained when adding the estimated DFV to the LA segmentation in phase 0 compare extremely well with the GT segmentations: $HD = 2.51 \pm 1.3$ $mm$; $DS = 0.96 \pm 0.02$. It outperformed previously used symmetric diffeomorphic image normalization from ANTs package (Avants et al., 2009) which obtained ($HD = 2.57 \pm 1.16$ $mm$; $DS = 0.85 \pm 0.04$) to the same LA contour images and ($HD = 3.35 \pm 1.48$ $mm$; $DS = 0.77 \pm 0.09$) when applied to the unsegmented LA images. Using LA-DNet directly on the unsegmented LA images as inputs also led to poor results ($HD = 3.35 \pm 1.05$ $mm$; $DS = 0.78 \pm 0.07$). The LA-DNet estimated DFVs are spatially and temporally smoother and the Jacobian of the deformation gradient is consistent with the known volumetric changes of the LA, as can be seen in: https://tinyurl.com/2eju3r9f.

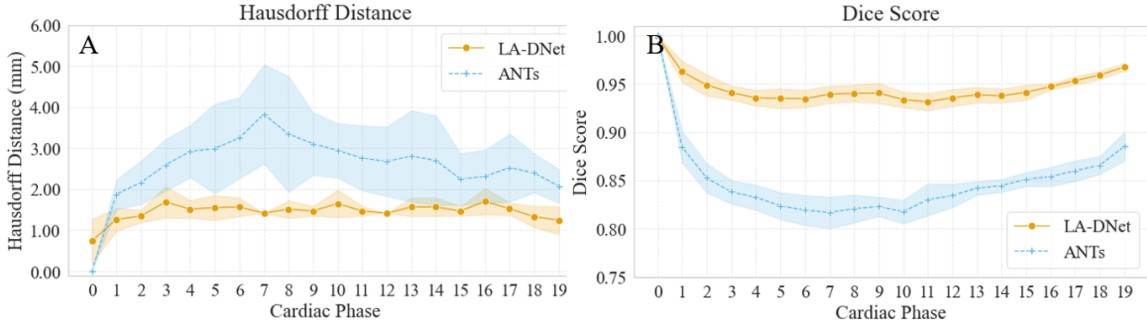

Figure 2: The image registration metrics plotted for LA-DNet and ANTs: A) Hausdorff distance (HD) and, B) Dice score (DS). HD and DS are obtained by comparing manual LA segmentations across the cardiac cycle with segmentations transformed using the estimated DFV on phase 0.

## 4. Conclusions

The proposed pipeline is able to extract DFVs that accurately track the LA wall across the cardiac cycle. The estimated high-resolution 3D LA DFVs pave the way towards potentially detecting regional functional biomarkers for conditions such as AF or HF. They may also provide useful information for the identification of LA fibrosis (Sohns and Marrouche, 2020).

The LA registration across the cardiac cycle is more challenging than that of the LV. For the latter, several registration tools are available (Hernandez et al., 2021; De Vos et al., 2019), but these performed poorly for the LA registration task. The usual assumption, that the intensity of the different image components (e.g. the LV myocardium) is constant across the cardiac cycle, is not valid for the LA. This is because the LA myocardium is very thin (Varela et al., 2017), and thus barely identifiable in bSSFP images; and the LA blood pool voxels' intensity depends on blood velocity and is therefore very variable across the cardiac cycle. We successfully propose a different approach for automatically LA registration, using LA contours from automated segmentations as inputs, training it on a subject by subject basis to allow its deployment to small datasets of Cine MRI of the LA.

## Acknowledgments

This work was supported by the UKRI CDT in AI for Healthcare http://ai4health.io (Grant No. EP/S023283/1) and the British Heart Foundation Centre of Research Excellence at Imperial College London (RE/18/4/34215). We acknowledge computational resources and support provided by the Imperial College Research Computing Service (http://doi.org/10.14469/hpc/2232). Last but not least, we thank the volunteers for allowing the use of their data for this research.

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
