# OpenReview forum: "High-resolution 3D Maps of Left Atrial Displacements using an Unsupervised Image Registration Neural Network"
_MIDL.io/2023/Short_Paper_Track — MIDL 2023 Short paper track Poster_

### Official Review · Reviewer_oMho · 2023-04-22
**Well-written paper with important clinical application**

**Rating:** 6
**Confidence:** 4

**Review:**

Summary:
The paper proposes a pipeline to automatically extract high-resolution 3D displacement field vector maps of the left atrium (LA) using a high-resolution cine MRI. The pipeline involves a segmentation network of the LA across the cardiac cycle to guide registration, extraction of the LA segmentation contour, and automatic subject-by-subject image registration of the LA contour image. The resulting displacement field vectors can potentially detect regional functional biomarkers for conditions such as atrial fibrillation or heart failure and identify LA fibrosis.

Strength:
1. The proposed pipeline shows promising results. The pipeline is able to accurately track the LA wall across the cardiac cycle with high precision.
2. The proposed method has a clear and important clinical application. The proposed method addresses a growing need for more precise biomarkers in the diagnosis and prognosis of cardiac conditions.
3. The paper is overall well-organized and the background introduction is clear and effectively establishes the need for the proposed pipeline. However, some details are missing with respect to the method.
4. It is good that the authors provide image registration metrics plotted for LA-DNet and ANTs across the cardiac cycle with multiple metrics, including the Hausdorf distance, Dice score, magnitude of displacement field vectors, and Jacobian determinant of DFV. Overall, Figure 2 demonstrates the strong performance of LA-DNet in accurately tracking the LA wall across the cardiac cycle.

Weakness:
1. Some details of training/registration strategy of LA-DNet are missing. The authors should clarify some details about the used loss function, input and output images of LA-DNet, etc.
2. More explanation is needed about "we parameterise them empirically" in section 2.3.  Please clarify how to empirically parameterize a neural network. More precisely, it should be the details of network architecture.

---

### Official Review · Reviewer_RFM1 · 2023-04-26
**An effective pipeline for LA segmentation and tracking**

**Rating:** 7
**Confidence:** 4

**Review:**

This paper proposes a tool dedicated to automatically segment the Left Artium (LA) and extracts the displacement fields across the cardiac cycle, using a novel high-resolution Cine MRI protocol designed specifically for the LA (with a spatial resolution that is higher than the existing datasets). The pipeline is able to
accurately track the LA wall across the cardiac cycle with a good average Hausdorff distance and Dice score .

As only a small amount of subjects have been imaged with the new protocol, the authors deployed a dedicated pipeline with 3 components: 1) a few-shot segmentation network across the cardiac cycle to guide registration; 2) extraction of the LA segmentation contour and dilation; and 3) a subject-by-subject image registration of the target contours.

The paper is clear and well written, and the experiments are well executed. This type of few-shot+registration approaches makes sense for the problem at hand, as the number of annotated samples is really limited. Therefore, I do not  see any issue with this submission and recommend acceptance.